# LIGHTNING VIDEO: BUILDING COMPACT DIFFUSION TRANSFORMERS FOR HIGH-FIDELITY ON-DEVICE VIDEO GENERATION

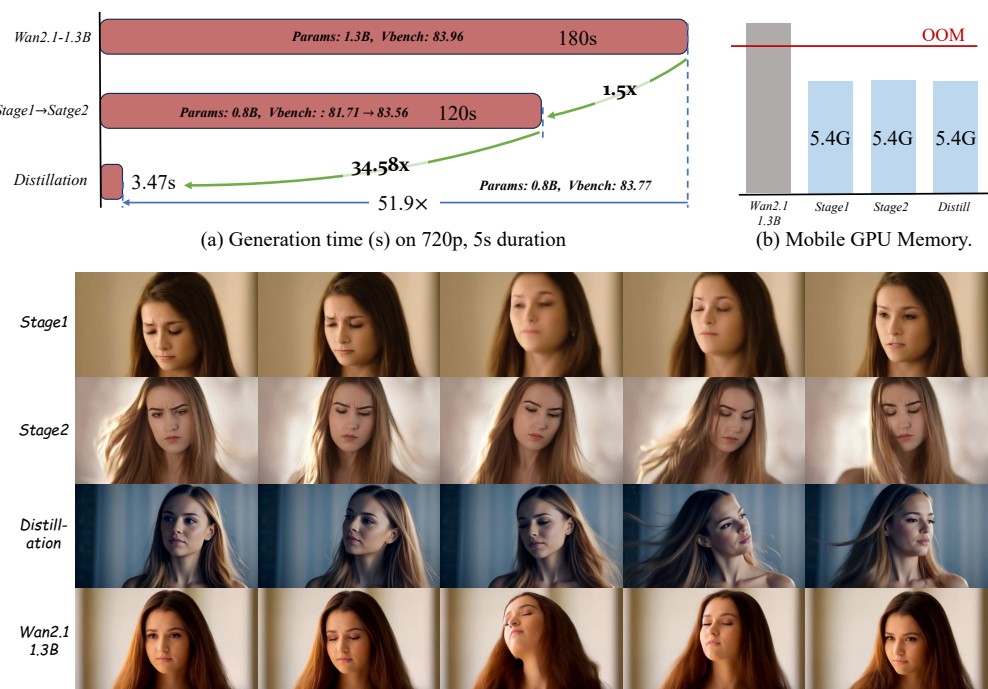

(a) Generation time (s) on 720p, 5s duration     (b) Mobile GPU Memory.

(c) Comparison of generated videos. Prompt: A person is shaking head

Figure 1: **Our method generates high-quality videos on mobile devices efficiently.** Our Lightning Video, a 0.8B compact diffusion transformer for video generation, achieves performance comparable to state-of-the-art open-sourced models. We report 720p, 5s duration video generation time on a single NVIDIA A800 GPU in (a), report its running time and memory usage on iPhone 16 Pro Max in (b) and compare the video generated by our model in different training stages in (c).

## ABSTRACT

Recent advances in Diffusion Transformers (DiTs) have enabled the generation of highly realistic video content, but state-of-the-art models often require billions of parameters, making them impractical for deployment on resource-constrained edge devices, such as smartphones. In this work, we introduce a systematic approach to designing lightweight yet powerful video DiTs tailored for edge scenarios. Our framework centers on three key components: (1) a Taylor-expansion–based pruning initialization that allows flexible model rescaling and rapid capability recovery with limited data; (2) a staged, data-efficient training protocol that couples this initialization with curated datasets and targeted optimization schedules; and (3) a distribution-matching distillation strategy that substantially reduces inference steps while preserving generation quality. We present **Lightning Video**, a 0.8B-parameter model that achieves competitive performance against billion-scale baselines while supporting native execution on edge devices (e.g., iPhone 16 Pro). These results demonstrate the feasibility of delivering high-quality video generation directly on end-user devices, opening new opportunities for practical mobile and creative applications.

# 1 INTRODUCTION

Recent advances in video generation (Henschel et al., 2025; Hong et al., 2022; Kong et al., 2024; Lin et al., 2024; Wan et al., 2025; Wang et al., 2025; Zheng et al., 2024; Ma et al., 2025; Gao et al., 2025; Brooks et al., 2024; Kuaishou) have demonstrated remarkable capabilities, producing movie-quality clips that are both realistic and rich in detail. Among existing approaches, diffusion transformers (DiTs) (Peebles & Xie, 2023) have emerged as the dominant paradigm, delivering state-of-the-art performance in terms of aesthetic quality, visual fidelity, and long-range temporal coherence. While such models highlight the potential of generative modeling for digital content creation, communication, and immersive media, their deployment on edge devices remains an open challenge. Native execution on smartphones and tablets would bring clear benefits—including low-latency generation, enhanced privacy, personalization, and ubiquitous accessibility—similar to the impact achieved by recent progress in on-device large language models. In the streaming era, enabling video generation directly on end-user devices could empower creative applications for a wide range of users.

However, existing state-of-the-art video models rely on billions of parameters and computationally expensive sampling schedules, making them infeasible for mobile deployment without aggressive quantization. Prior research on mobile visual generation either focuses on optimizing individual components (Zou et al., 2025; Wu et al., 2025b) or relies primarily on U-Net–based network architectures (Wu et al., 2025c; Chen et al., 2025; Yahia et al., 2024; Zhao et al., 2024; Zhang et al., 2024b), however, they typically yield compromised generation quality and diverge from the prevailing diffusion transformer paradigm. This architectural deviation limits their representational power and hinders their ability to leverage widespread advancements such as LoRA-based parameter-efficient adaptation. In contrast, attempts to develop mobile-friendly DiT video generation models remain scarce. Recent efforts (Wu et al., 2025a) adhere to a multi-stage pipeline of pruning, training, and step distillation. While its structural pruning stage induces a significant performance degradation compared to the base model and its training strategies require further improvement. Consequently, its generation performance continues to lag substantially behind that of state-of-the-art systems.

In this work, we present a general methodology for constructing compact yet powerful Diffusion Transformer models for video generation, as shown in Figure 1. Our approach is built on two key ideas. First, we introduce an effective and efficient initialization scheme based on Taylor-expansion pruning with no requirement of backward process and massive data, which enables arbitrary model resizing while retaining expressive capacity and allows rapid recovery of generative ability from minimal data. Second, we design a staged training protocol that leverages priors and modest yet high-quality datasets, yielding fast convergence in compact models without requiring massive-scale data. To further enhance inference efficiency, we conduct a distribution matching distillation post-training that compresses diffusion sampling into only a few steps, making real-time generation feasible on mobile hardware.

With these innovations, we present a 0.8B parameter DiT model that achieves performance comparable to models with 2B–8B parameters, while maintaining quantization-free, on-device execution. This result demonstrates that compact diffusion models, when paired with principled initialization and staged training, can rival the performance of much larger systems at a fraction of the compute cost.

Our contributions can be summarized as follows:

- We propose a unified methodology for building compact Diffusion Transformers tailored for resources-constrained environment applications, such as on-device video generation.

- We introduce a computationally efficient Taylor–expansion–based initialization, supporting flexible model scaling and fast capability recovery.

- We develop a staged training framework that integrates strong priors with high-quality, modest-sized datasets for rapid convergence in small models.

- We conduct a distribution matching distillation method that reduces diffusion sampling steps and improves on-device inference efficiency of the model further. We demonstrate that a 0.8B parameter model achieves results comparable to 2B–8B baselines and release all code, pipelines, and pretrained weights to support open research.

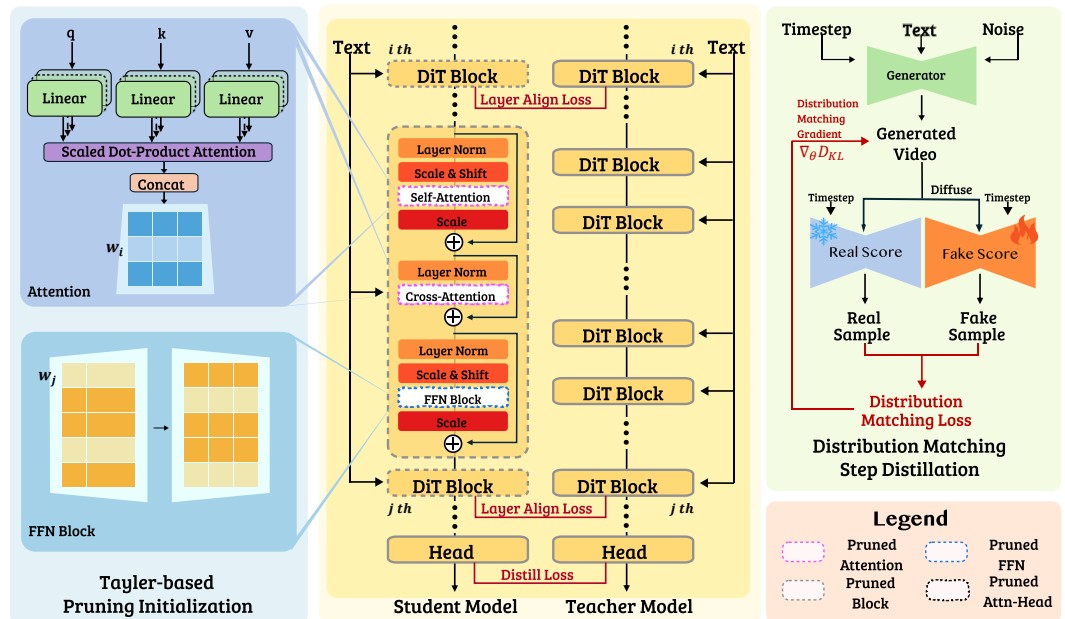

Figure 2: **Architecture of Lightning Video and our three stages framework to build the model.** The figure demonstrates our pruning initialization, training then step distillation framework to build a compact diffusion transformer.

## 2 METHODS

### 2.1 TAYLER-BASED PRUNING INITIALIZATION

Training small yet competitive models often suffers from slow convergence and unstable optimization due to limited parameter capacity. We found that an effective initialization strategy plays a key role in mitigating the training challenges described above. Existing methods (Fang et al., 2023; 2025) either have not proved their effectiveness on the flow matching DiT model or require a costly backwards process to compute the importance scores. We propose a Taylor-based sensitivity analysis method that enables weight initialization through a principled pruning perspective, which only requires model inference and is simple and effective. The procedure is shown on the left side of Figure 2.

Concretely, consider a linear mapping in feed-forward networks (FFN) or in the projection matrices of attention modules in the DiT network, where the transformation can be defined as $\hat{\mathbf{Y}} = \mathbf{X}\mathbf{W}^T + \mathbf{1}\boldsymbol{b}$. The prediction error with respect to the target output of this linear layer $\mathbf{Y}$ is the difference between the predicted and target outputs, which can be expressed as:

$$\mathbf{E} = \hat{\mathbf{Y}} - \mathbf{Y} \tag{1}$$

where $\mathbf{X} \in \mathbb{R}^{n \times d_{in}}$ is the input feature, $\mathbf{W} \in \mathbb{R}^{d_{out} \times d_{in}}$ and $\boldsymbol{b} \in \mathbb{R}^{1 \times d_{out}}$ represent the parameter matrix and bias array, respectively, and $\mathbf{1} \in \mathbb{R}^{n \times 1}$ is an all-ones vector for broadcasting the bias. The loss function $L$ is defined as the squared Frobenius norm of the error matrix, which is equivalent to the sum of squared errors and can be computed using the trace operator:

$$L \propto ||\mathbf{E}||_F^2 = \text{tr}(\mathbf{E}^T\mathbf{E}) = \text{tr}\left((\mathbf{X}\mathbf{W}^T + \mathbf{1}\boldsymbol{b} - \mathbf{Y})^T(\mathbf{X}\mathbf{W}^T + \mathbf{1}\boldsymbol{b} - \mathbf{Y})\right) \tag{2}$$

We only consider the row-wise dimensions of the weight matrix, which correspond to the feed-forward network dimension (ffn_dim) and the number of attention heads. Since these dimensions directly determine the output neurons or filters, their contribution to the loss must be carefully quantified. We therefore formulate the parameter vector as $\boldsymbol{w}_i$ representing the weights associated with the $i$-th row of the matrix. To quantify the importance $I_i$ of the $i$-th row, we approximate the loss variation $\Delta L$ induced by pruning this neuron via a second-order Taylor expansion around a stationary point where the gradient vanishes:

$$I_i = \Delta L_i \approx \frac{\partial L}{\partial \boldsymbol{w_i}}\boldsymbol{w_i} + \frac{1}{2}\boldsymbol{w_i}^\top \mathbf{H}\boldsymbol{w_i} \approx \boldsymbol{w_i}^\top \boldsymbol{x}^\top \boldsymbol{x}\boldsymbol{w_i} \tag{3}$$

where $\mathbf{H}$ denotes the Hessian block corresponding to the second-order derivatives of the loss with respect to the parameters in the $i$-th row. Since the base model has already converged on the training dataset, we neglect the first-order term where $\partial L / \partial \boldsymbol{w}_i \approx 0$. After evaluating the derivatives in equation 3, we obtain the final expression for the saliency score of the $i$-th row. Moreover, since each row is associated with both weights and a bias term, the pruning rule for the bias $b_i$ follows the same formulation as for $\boldsymbol{w}_i$, and its saliency score is computed analogously. A step-by-step derivation is provided in Appendix B. In our experiments, we keep the model depth and hidden dimensions unchanged while pruning the attention heads from 12 to 10 and reducing the FFN intermediate dimensions from 8960 to 4480, resulting in a model of 850M parameters.

## 2.2 Multi-stage Training

To effectively build a compact yet strong video model upon our initialized model with a relatively low training budget, we adopt a two-stage training strategy using high-quality image-video mixed data. In the first stage, we choose Wan2.1-1.3B-T2V (Wan et al., 2025) as the teacher, and the model is jointly optimized with the flow matching loss and distillation objectives, which include output-level and intermediate-layer alignment. In the second stage, we remove the distillation terms and continue optimizing the model only with flow matching, which allows the student to further refine generation quality.

**Stage I: Flow Matching with Multi-level Alignment.** Flow matching defines a continuous-time generative process by transporting a base distribution $p_0(x)$ (e.g., Gaussian noise) to the target data distribution $p_1(x)$. The key idea is to learn a vector field $v_\theta(x, t)$ such that the probability flow ODE: $\frac{dx_t}{dt} = v_\theta(x_t, t)$ transports $x_t$ to $p_1$ at $t = 1$, where $x_t = (1 - t)x_0 + tx_1$ is the linear interpolation between data and noise and $x_0 \sim p_0$. The flow matching model training objective can be written as:

$$\mathcal{L}_{\text{FM}} = \mathbb{E}_{t \sim \mathcal{U}(0,1),\, (x_0, x_1)} \left[ \left\| v_\theta(x_t, t) - \frac{x_1 - x_0}{t} \right\|_2^2 \right], \tag{4}$$

To ensure efficient knowledge transfer from the teacher model $f_T$ to the student $f_\theta$ and to accelerate the convergence of the initialized model, we introduce two additional distillation losses. First, we apply an output alignment loss, which encourages the student's output to match that of the teacher at the final layer, defined as:

$$\mathcal{L}_{\text{output}} = \mathbb{E}_{x \sim \mathcal{D}} \left[ \| f_\theta(x) - f_T(x) \|_2^2 \right]. \tag{5}$$

Previous works (Ren et al., 2025; Yu et al., 2024; Jiang et al., 2025) inspire us that aligning intermediate features between teacher and student can further improve training efficiency. Thus, we align selected intermediate representations $\{h_\theta^l(x)\}$ of the student with their teacher counterparts $\{h_T^l(x)\}$:

$$\mathcal{L}_{\text{layer-align}} = \sum_{l \in \mathcal{S}} \mathbb{E}_{x \sim \mathcal{D}} \left[ \left\| h_\theta^l(x) - h_T^l(x) \right\|_2^2 \right], \tag{6}$$

where $\mathcal{S}$ denotes the set of aligned layers. Empirically validated, we adopt the L2-norm loss. We find it simple and can further stabilize and improve optimization without introducing additional learnable parameters. As the middle part of Figure 2 shows, the overall objective in Stage I is therefore expressed as:

$$\mathcal{L}_{\text{Stage I}} = \mathcal{L}_{\text{FM}} + \lambda_{\text{output}} \mathcal{L}_{\text{output}} + \lambda_{\text{layer-align}} \mathcal{L}_{\text{layer-align}}, \tag{7}$$

where $\lambda_{\text{out}}$ and $\lambda_{\text{layer}}$ are weighting coefficients. $\lambda_{\text{out}}$ and $\lambda_{\text{layer}}$ are set to 1.0 in our experiments.

**Stage II: Fine-tuning.** Once the student has acquired sufficient knowledge from the teacher, the distillation objectives are removed, and training proceeds only with the flow matching loss:

$$\mathcal{L}_{\text{Stage II}} = \mathcal{L}_{\text{FM}}. \tag{8}$$

At this stage, the student model is no longer constrained to overfit the teacher but instead learns to refine its generative capability independently. This design is motivated by our experimental finding that a synergistic stage-gating between alignment and fine-tuning yields superior performance. Specifically, alignment accelerates initial convergence in the first stage, while fine-tuning drives optimal performance gains in the second stage. Detailed analysis can be found in Section 3.3

## 2.3 Distribution Matching Step Distillation

To further reduce the inference cost of our compact video generation model and make it practical for deployment on mobile devices with acceptable latency, we adopt step distillation based on the improved Distribution Matching Distillation framework (Yin et al., 2024b;a). As the right part of Figure 2 presents, let $G_\theta$ denote the student generator with parameters $\theta$, $z \sim \mathcal{N}(0, I)$ be the input noise, $t'$ is sampled from the student generator schedule, and $z_{t'}$ is retrieved by simulating the denoising process using the student to $t'$. We denote by $F(\cdot, t)$ the forward diffusion operator that injects noise corresponding to timestep $t \sim \mathcal{U}(0, T)$. Following Yin et al. (2024a), the gradient of distribution matching loss $\mathcal{L}_{\text{Distill}}$ over few steps generator can be written as

$$\nabla_\theta \mathcal{L}_{\text{Distill}} = \mathbb{E}_{t,t',z} \left[ \left( s_{\text{real}}(F(G_\theta(z_{t'}), t), t) - s_{\text{fake}}(F(G_\theta(z_{t'}), t), t) \right) \frac{\partial G_\theta(z_{t'})}{\partial \theta} \right], \tag{9}$$

where $s_{\text{real}}$ and $s_{\text{fake}}$ are score functions (Song et al., 2020) estimated respectively from a frozen diffusion model and a learnable critic trained on the student outputs.

In our setting, the distilled student operates with a small number of denoising steps (typically $N = 4$), significantly lowering inference complexity while preserving visual fidelity. The final loss for our distillation stage is therefore where training directly minimizes the distribution discrepancy without auxiliary regression or adversarial objectives. This design allows the compact student model to maintain high-quality synthesis while being deployable on resource-constrained edge devices.

## 3 Experiments

**Data construction.** For the image dataset, we collected approximately 5M high-quality text–image pairs from Lumina-Image 2.0 (Qin et al., 2025) and Lumina-mGPT 2.0 (Xin et al., 2025), which are leveraged to facilitate joint training across image and video modalities. For the video dataset, we curated 1.5M samples from several publicly available datasets, including Open-Sora-Plan (Lin et al., 2024), OpenHumanVid (Li et al., 2025), OpenVid-1M (Nan et al., 2024), and Panda-10M (Chen et al., 2024b), etc, providing a broad spectrum of human activities and real-world scenarios. In addition, we constructed a supplementary dataset by collecting 3.5M movie clips (segmented into 5-second snippets) and crawling videos from the web. For both the first and second training stages, we adopt a joint image–video training paradigm. The main difference is that, in the second stage, we manually curate a subset of high-quality videos and use them exclusively for fine-tuning. During the distillation stage, only text prompts are provided as input.

**Training setup.** We use the Wan2.1-1.3B-T2V model (Wan et al., 2025) as our base model for the initialization process, which costs about 5 minutes for inference on a single GPU. The Stage I alignment is conducted on 32 NVIDIA A800 80G GPUs for $10K$ iterations using the AdamW optimizer (Loshchilov & Hutter, 2017) with a learning rate of $1e - 4$ and beta values of $[0.9, 0.999]$. We gradually reduce the learning rate from $1e - 4$ to $5e - 5$ in the Stage II fine-tuning process, which is conducted for $10K$ iterations on 64 NVIDIA A800 80G GPUs. We set beta values as $[0.9, 0.95]$, weight decay as 0, and eps as $1e - 15$ for better convergence in Stage II. Stage I and Stage II progress use high quality open-sourced as well as internal collected image and video data.

**Distillation setup.** Following previous works, we collect $300K$ high quality prompts for distribution matching distillation stage. The distillation experiment is conducted on 32 NVIDIA A800 80G GPUs for $4K$ iterations. AdamW optimizer is adopted with a learning rate of $1e - 6$ and beta values of $[0.0, 0.999]$.

**Evaluation setup.** All evaluation results of our models follow the official procedure of widely-used VBench (Huang et al., 2024). Following CogVideoX (Yang et al., 2024), we use the longer prompt rewritten by GPT-4o (Hurst et al., 2024) from VBench official repository and generate 4 samples for each prompt using different random seeds for all our experiments.

**Deployment on iPhone 16 pro max.** We deploy our few step model on the iPhone 16 pro max with 8 GB unified memory. We use an INT4 quantized version of the UMT5-xxl-encoder model as the text encoder to avoid OOM on the iPhone. Wan VAE decoding is the main speed bottleneck of

Table 1: **Performance comparison with popular video generation models on VBench.** Our proposed methods (highlighted in green) achieve competitive performance with significantly fewer parameters.* denotes support for running on mobile devices.

| Model | Params (B) | Steps | Total score | Quality score | Semantic score | aesthetic quality | motion smoothness | dynamic degree | object class | spatial relationship |
|---|---|---|---|---|---|---|---|---|---|---|
| **Closed-Source Models** | | | | | | | | | | |
| **SnapDiT-Server(Wu et al., 2025a)** | 2.0 | – | 83.09 | 84.65 | 76.86 | 64.72 | – | – | 90.57 | – |
| **Kling-2407-High(Kuaishou)** | – | – | 81.85 | 83.39 | 75.68 | 61.21 | 99.40 | 46.94 | 87.24 | 73.03 |
| **SnapDiT-Mobile*(Wu et al., 2025a)** | 0.9 | 4 | 81.45 | 83.12 | 74.76 | 64.16 | – | – | 92.26 | – |
| **SnapGen-V*(Wu et al., 2025c)** | 0.6 | 4 | 81.14 | 83.47 | 71.84 | 62.19 | 99.29 | 51.11 | 92.22 | 56.20 |
| **Pika-1.0(Pika AI)** | – | – | 80.69 | 82.92 | 71.77 | 62.04 | 99.50 | 47.50 | 88.72 | 61.03 |
| **ModelScope(Modelscope AI)** | 1.4 | 50 | 75.75 | 78.05 | 66.54 | 52.06 | 95.79 | 66.39 | 82.25 | 33.68 |
| **Open-Source Models** | | | | | | | | | | |
| **Wan2.1-14B(2503)(Wan et al., 2025)** | 14 | 50 | 86.22 | 86.67 | 84.44 | – | – | – | – | – |
| **Wan2.1-1.3B(2503)(Wan et al., 2025)** | 1.3 | 50 | 83.96 | 84.92 | 80.10 | – | – | – | – | – |
| **HunyuanVideo(2412)(Kong et al., 2024)** | 14 | 50 | 83.24 | 85.09 | 75.82 | 60.36 | 98.99 | 70.83 | 86.10 | 68.68 |
| **CogVideoX 1.5(Yang et al., 2024)** | 5 | 50 | 82.17 | 82.78 | 79.76 | 62.79 | 98.31 | 50.93 | 83.42 | 80.25 |
| **Pyramid Flow(Lei et al., 2023)** | 2.0 | 20 | 81.72 | 84.74 | 69.62 | 63.26 | 99.12 | 64.63 | 86.67 | 59.53 |
| **CogVideoX-5B(Yang et al., 2024)** | 5.0 | 50 | 81.61 | 82.75 | 77.04 | 61.98 | 96.92 | 70.97 | 85.23 | 66.35 |
| **T2V-Turbo(Li et al., 2024)** | 1.6 | 4 | 81.01 | 82.57 | 74.76 | 63.04 | 97.34 | 49.17 | 93.96 | 38.67 |
| **Emu3(Wang et al., 2024c)** | 8.0 | – | 80.96 | 84.09 | 68.43 | 59.64 | 98.93 | 79.27 | 86.17 | 68.73 |
| **CogVideoX-1.6B(Yang et al., 2024)** | 1.6 | 50 | 80.91 | 82.18 | 75.83 | 60.82 | 97.73 | 59.86 | 87.37 | 69.90 |
| **VideoCrafter-2.0(Chen et al., 2024a)** | 1.9 | 50 | 80.44 | 82.20 | 73.42 | 63.13 | 97.73 | 42.50 | 92.55 | 35.86 |
| **AnimateDiff-V2(Guo et al., 2023)** | 1.2 | 25 | 80.27 | 82.90 | 69.75 | 67.16 | 97.76 | 40.83 | 90.90 | 34.60 |
| **OpenSora V1.2(Zheng et al., 2024)** | 1.2 | 30 | 79.76 | 81.35 | 73.39 | 56.85 | 98.50 | 42.39 | 82.22 | 68.56 |
| **AnimateLCM(Wang et al., 2024b)** | 1.2 | 4 | 79.42 | 82.36 | 67.65 | 67.01 | 98.16 | 40.56 | 91.41 | 37.14 |
| **Ours multi-steps*** | 0.8 | 50 | 83.56 | 84.66 | 79.18 | 66.47 | 97.53 | 77.78 | 93.47 | 71.33 |
| **Ours few-steps*** | 0.8 | 4 | 83.77 | 85.39 | 77.29 | 66.35 | 98.50 | 81.94 | 94.78 | 74.87 |

the entire video generation pipeline. (Zou et al., 2025) To accelerate the decoding process, we use TAE-wan2.1, making it 10 times faster compared to the original Wan2.1 VAE. All components are implemented in the Apple-developed *mlx-swift* framework (Hannun et al., 2023), providing a flexible environment for model customization during development.

### 3.1 QUANTITATIVE EVALUATION

We present comprehensive quantitative comparison between our models and popular academic and commercial video generation models on VBench (Huang et al., 2024), as shown in Table 1. We generate 81-frame horizontal videos at a resolution of $480 \times 832$ for our multi-steps and few-steps model, which are saved at 5 seconds(16 FPS) for quantitative evaluation qualitative comparison. The results demonstrate that our method achieves performance comparable to many large-scale models while requiring significantly fewer parameters.

In particular, our few-steps model, with only 0.8B parameters and 3 inference steps, achieves a total score of 83.77, which is on par with or superior to several larger models such as Pika-1.0 (Pika AI) (80.69, parameters not disclosed) and ModelScope (Modelscope AI) (75.75, 1.4B). Moreover, our approach substantially outperforms other lightweight open-source models (e.g., AnimateLCM(Wang et al., 2024b) with 1.2B parameters and 4 steps, scoring 79.42). This highlights the effectiveness of our design in balancing efficiency and quality. Notably, our models also demonstrate strong results across multiple fine-grained metrics. For instance, both variants yield higher scores in object class score (above 93) compared to most competing methods, while maintaining competitive spatial relationship scores.

An additional advantage of our method supports mobile device, as denoted by "*" in Table 1. Compared to other models that can run on mobile devices, such as SnapGen-V(Wu et al., 2025c) and SnapDiT-Mobile(Wu et al., 2025a), our approach achieves superior performance, e.g., total score.

### 3.2 VISUALIZATION COMPARSION

We compare the generated videos from our multi-steps model and few-steps model with Wan2.1-1.3B-T2V in Fig 3. As shown, our approach achieves higher realism and visual quality, e.g., the cat in (b) and the red panda playing the guitar in (c). The Ours-fewsteps variant demonstrates better shape

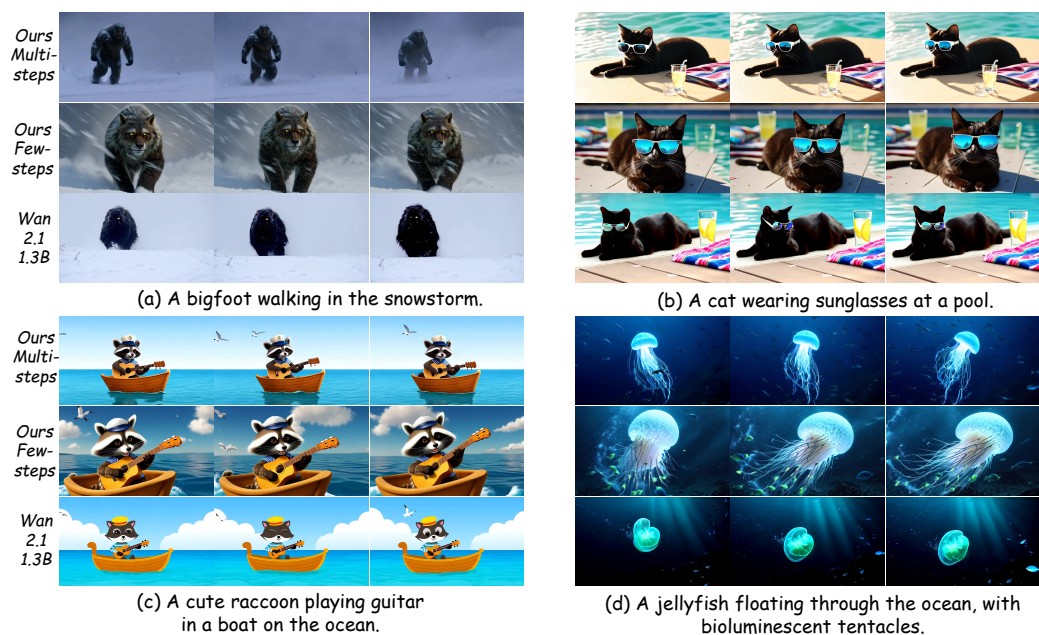

(a) A bigfoot walking in the snowstorm.

(b) A cat wearing sunglasses at a pool.

(c) A cute raccoon playing guitar
in a boat on the ocean.

(d) A jellyfish floating through the ocean, with
bioluminescent tentacles.

Figure 3: Visualization comparison of Ours (Multi-steps), Ours (Few-steps), and Wan 2.1-1.3B.

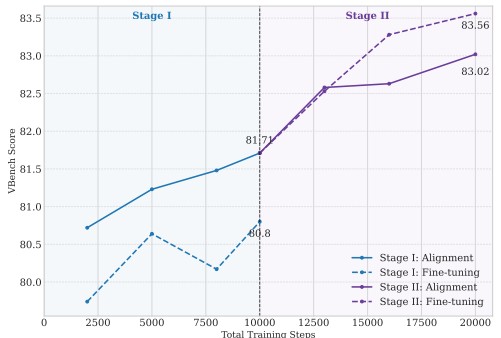

Figure 4: **Training strategy analysis.** VBench score evolution versus training steps, comparing different training strategies across stages.

Table 2: **Stage I training loss effectiveness.** VBench (Huang et al., 2024) scores with different loss combinations in distillation training.

| Stage I Strategy | MO | SR | AQ | Quality | Semantic | Total |
|---|---|---|---|---|---|---|
| w/o layer-align | 61.13 | 71.99 | 62.66 | **82.91** | 74.36 | 81.20 |
| w/o image data | 58.31 | 62.56 | 61.42 | 82.54 | 71.85 | 80.40 |
| Full | **65.24** | **75.20** | **64.07** | 82.82 | **77.26** | **81.71** |

Table 3: **Stage II training scheme effectiveness.** VBench (Huang et al., 2024) scores with different strategies in Stage II training.

| Stage II Strategy | AS | DD | OC | Quality | Semantic | Total |
|---|---|---|---|---|---|---|
| Alignment | 22.89 | 62.50 | 26.94 | 83.66 | **80.50** | 83.02 |
| Fine-tuning | **23.06** | **77.78** | **27.00** | **84.66** | 79.18 | **83.56** |

consistency, such as the bigfoot in (a), whereas Wan2.1-1.3B fails to produce recognizable faces. Furthermore, after the distillation step, we observe a clear improvement in visual fidelity, validating the effectiveness of this stage.

## 3.3 ABLATION STUDY

We analyze the impact of the Stage I loss and the overall training scheme, and further present and discuss detailed on-device performance results. More results and discussion can be found in Appendix C.

**Effect of Training Scheme in Stage I.** To better demonstrate the effectiveness of the training losses used in the alignment stage, we calculate the VBench score after removing the layer-wise alignment loss. We also evaluate the influence of the image–video mixed training strategy. All the models are trained for $10K$ iterations under Stage I setting. As shown in Table 2, incorporating image data leads to notable performance improvements across all aspects. Moreover, the layer-wise alignment loss significantly improves the model's aesthetic quality as well as its overall performance.

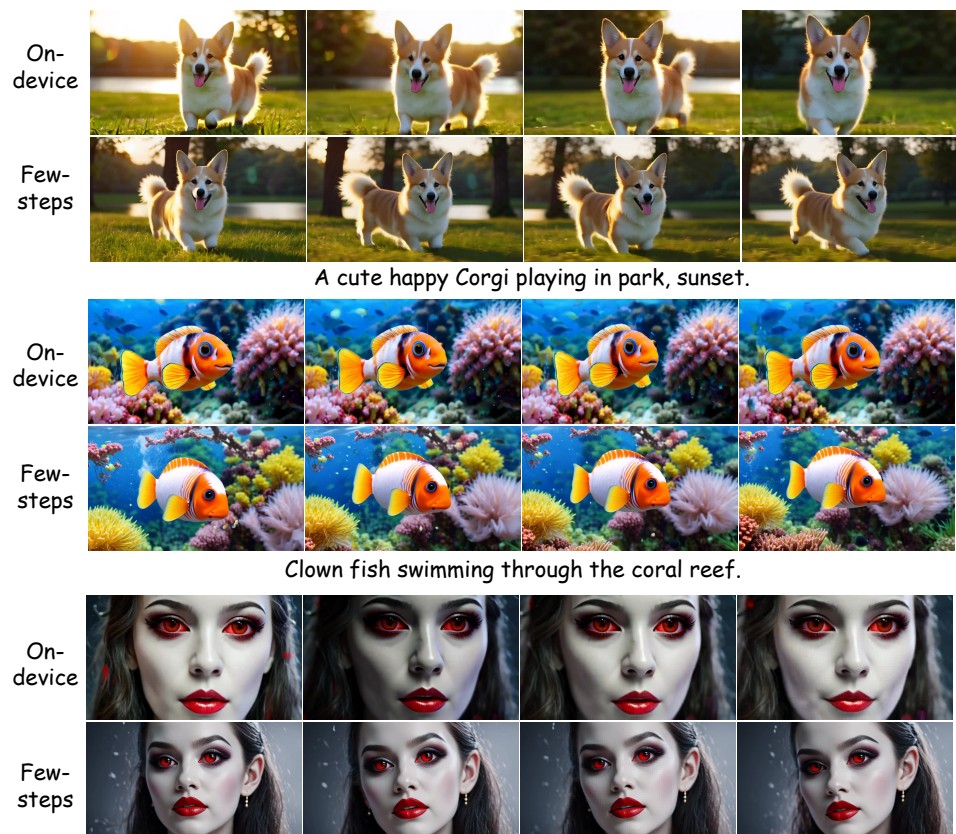

On-device

Few-steps

A cute happy Corgi playing in park, sunset.

On-device

Few-steps

Clown fish swimming through the coral reef.

On-device

Few-steps

Vampire makeup face of beautiful girl, red contact lenses.

Figure 5: Visualization comparing on-device deployment and an A800 GPU server.

**Analysis of alignment and fine-tuning training strategy.** We analyze the training strategies in Stage I (alignment) and Stage II (fine-tuning) to identify the optimal approach for performance and convergence speed. First, we start from the Stage I alignment checkpoint and compare the two strategies using the same Stage II data, with results presented in Table 3. The table shows that the model trained with the fine-tuning strategy achieves a higher final VBench score (83.56 vs. 83.02) and demonstrates significantly better quality and dynamic degree scores.

We hypothesize that this is due to a conflict between the alignment gradient from the teacher model and the training gradient from the real video data. This conflict likely arises from the domain gap between our training data and the teacher model's pre-training data, which prevents the model from achieving its highest potential performance.

Figure 4 further illustrates this dynamic. In the initial phase, the alignment strategy leverages knowledge distillation, leading to faster initial convergence, while the fine-tuning strategy starts more slowly due to the absence of teacher guidance. However, upon transitioning to Stage II, the unconstrained fine-tuning strategy demonstrates a more effective and sustained performance gain, ultimately surpassing the alignment strategy to reach a higher peak.

**Device Performance Analysis.** Generating videos in the resolution of 480×832 costs amount of time on device. We decrease the resolution and frames of generated videos in our demo. The detailed performance of our few-steps model on iPhone 16 pro max is reported in Table 4. We generate videos using our few-steps model under on-device setting with TAE-wan2.1 and quantized INT4 text encoder. All videos are generated on mobile device and evaluated on GPU. From Table 4, we observe that our method achieves a multi-fold speedup on mobile devices, though there remains gap (≈10 fps) compared to other models such as SnapGen-V(Wu et al., 2025c). This limitation mainly stems from the higher compression ratio of the VAE we adopt (4×8×8), in contrast to SnapGenV's 4×16×16 setting. In addition, Fig. 5 presents a visual comparison between on-device and server-side inference, showing that the video quality on mobile devices is nearly indistinguishable from that on the server.

Table 4: **On-device deployment performance.**

| Setting | Resolution | Frames | Infer-Time (s) | Quality | Semantic | Total |
|---|---|---|---|---|---|---|
| **Few-steps full** | 480×832 | 81 | 598 | 85.39 | 77.29 | 83.77 |
| **Few-steps device** | 352×624 | 49 | 73 | 85.25 | 72.53 | 82.72 |

## 4 RELATED WORKS

**Video Generation Models.** Recent advances in generative modeling have substantially improved text-to-video (T2V) systems. Leading models such as Sora (Brooks et al., 2024), MovieGen (Polyak et al., 2024), and Veo 3 (Google Deepmind) exemplify this shift, integrating high-resolution latent representations, dense captioning modules, and sophisticated cross-frame attention mechanisms to produce temporally coherent, high-fidelity videos with flexible compositionality. In parallel, open-source contributions have accelerated architectural innovation and accessibility. HunyuanVideo (Kong et al., 2024) leverages a 13B-parameter multimodal LLM to enable high-quality generation across diverse prompts. CogVideoX (Yang et al., 2024; Hong et al., 2022) and LTX-Video (HaCohen et al., 2024) adopt unified transformer backbones with improved VAE-based spatial compression to enhance fidelity and efficiency. The Wan2 series (Wan et al., 2025) further extends this line through spatiotemporal VAEs, DiT-based generation blocks, and Mixture-of-Experts routing, supporting efficient synthesis of 480p–720p videos at 24fps on a single GPU and bridging the gap between quality and deployment constraints.

**Diffusion Accelerate.** To meet the growing demand for real-time and low-latency synthesis, a large body of work compresses diffusion sampling into few- or even one-step generators. These include distillation-based approaches (e.g., Distribution-Matching Distillation (DMD) (Yin et al., 2024c), DMD2 (Yin et al., 2024a)), consistency-based frameworks (Song et al., 2023; Lu & Song, 2024; Wang et al., 2024a), and alternative formulations such as Inductive Moment Matching (Zhou et al., 2025), Shortcut Models (Frans et al., 2024), and MeanFlow (Geng et al., 2025). In parallel, adversarial post-training methods (e.g., LADD (Sauer et al., 2024), APT (Lin et al., 2025)) combine teacher guidance with adversarial objectives for improved one-step generation. These methods are orthogonal to speed-oriented schedulers and pruning strategies. Approaches such as Diff-Pruning Fang et al. (2023) primarily target U-Net–based DDPM pruning, whereas others Fang et al. (2025); Zhang et al. (2024a); Castells et al. (2024) investigate how to preserve or recover model capacity as mush as possible. However, these techniques have been validated only on image generation tasks and neglect the challenges of large-scale, open-domain video generation.

**On-Device Video Generation.** Simultaneously, on-device video generation has attracted growing attention, driven by the need for privacy, low latency, and interactivity in real-world deployment. Wu et al. (2025a) propose a compression-aware adaptation of Diffusion Transformers, combining lightweight VAEs, tri-level pruning, and adversarial step distillation to achieve real-time synthesis on iPhone 16 Pro Max at over 10 FPS with only four denoising steps. On-device Sora (Kim et al., 2025) adopts a training-free acceleration strategy to optimize model layer and inference process, enabling efficient diffusion inference on iPhone 15 Pro. In parallel, SnapGen-V (Wu et al., 2025c) introduces a lightweight U-Net-based denoising architecture optimized for temporal modeling, achieving generation of 5-second videos within 5 seconds on mobile devices. However, the visual quality of current mobile systems remains limited.

## 5 CONCLUSION

We introduce a framework for constructing compact yet powerful diffusion transformers for video generation on edge devices. By combining Taylor-expansion–based pruning initialization, a staged data-efficient training protocol, and distribution-matching distillation, our approach enables flexible model size compression, efficient training, and efficient inference while preserving generation quality. The resulting Lightning Video, with only 0.8B parameters, is on par with multi-billion-parameter baselines and executes on mobile hardware without quantization. These results establish the feasibility of high-quality video generation directly on end-user devices, paving the way for practical and accessible mobile creativity.

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

## A  LLM USAGE

The authors used Large Language Models (LLMs), including ChatGPT, Gemini, and Grok, as editing assistants to polish the writing and identify errors in the paper. All content was initially written by the authors, and LLMs' suggestions were then incorporated during the revision process.

## B  DETAILED ANALYSIS OF TAYLOR-BASED INITIALIZATION

Here we demonstrate detailed derivation process of result in Equation 3 We start from the squared-error loss of a linear transformation,

$$
\begin{aligned}
L = \mathrm{tr}(EE^\top), \quad E &= XW^\top - Y \\
&= \mathrm{tr}\left[(XW^\top - Y)(WX^\top - Y^\top)\right] \\
&= \mathrm{tr}\left[XW^\top WX^\top - YWX^\top - XW^\top Y^\top + YY^\top\right] \\
&= \mathrm{tr}\left[X\left(\sum_{i=1}^c w_i^\top w_i\right)X^\top - YWX^\top - XW^\top Y^\top + YY^\top\right] \\
&= \mathrm{tr}\left[\sum_{i=1}^c X(w_i^\top w_i)X^\top - YWX^\top - XW^\top Y^\top + YY^\top\right]
\end{aligned}
\tag{10}
$$

where $W = [w_1; \ldots; w_c]$ stacks row vectors $w_i \in \mathbb{R}^{d_{in}}$. From equation 10, the contribution of each row can be isolated, and the Hessian block associated with $w_i$ is given by

$$
\frac{\partial^2 L}{\partial w_i \partial w_i^\top} = 2X^\top X.
\tag{11}
$$

To assess the saliency of the $i$-th neuron, we consider pruning it by perturbation $\Delta w_i = -w_i$. Applying a second-order Taylor expansion of the loss around a stationary point (where $\partial L / \partial w_i \approx 0$) yields

$$
\Delta L_i \approx \frac{1}{2} \Delta w_i^\top \left(\frac{\partial^2 L}{\partial w_i \partial w_i^\top}\right) \Delta w_i = \frac{1}{2}(-w_i)^\top (2X^\top X)(-w_i) = w_i^\top X^\top X w_i.
\tag{12}
$$

Thus, the row-wise saliency score used in the main text takes the closed form

$$
I_i = \Delta L_i \approx w_i^\top X^\top X w_i.
\tag{13}
$$

## C  MORE EXPERIMENT RESULTS

### C.1  TAYLOR-BASED SENSITIVITY PRUNING INITIALIZATION

Table 5: **Comparison of learnable masked initialization strategy and our taylor-based pruning strategy.** We measure VBench (Huang et al., 2024) score of model trained with Stage I setting using learnable masked initialization strategy and our taylor-based pruning strategy. In the results, "MO", "SR", and "AQ" denote the multiple objects, spatial relationship, and aesthetic quality scores, respectively. The best results are highlighted in bold.

| # Initialization Strategy | MO | SR | AQ | Quality | Semantic | Total |
|---|---|---|---|---|---|---|
| **Learnable masked** | 52.90 | 68.06 | 62.05 | 82.08 | 70.98 | 79.86 |
| **Ours** | **65.24** | **75.20** | **64.07** | **82.82** | **77.26** | **81.71** |

We provide additional experimental results to demonstrate the effectiveness of our model initialization methods. Without initialization, the model fails to generate valid videos after distillation or training using our data, as training long video clips from scratch is highly challenging. Furthermore, we

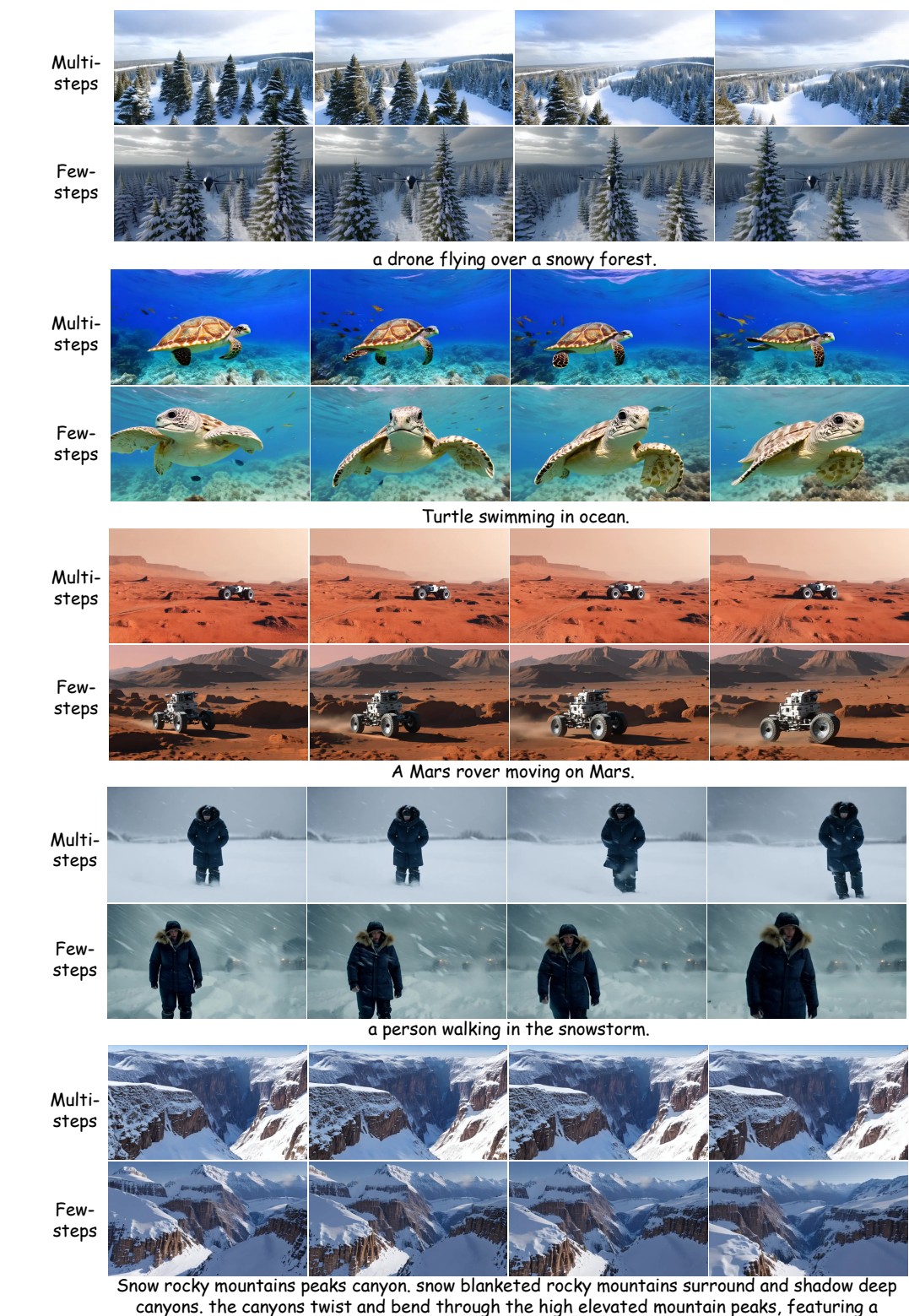

a drone flying over a snowy forest.

Turtle swimming in ocean.

A Mars rover moving on Mars.

a person walking in the snowstorm.

Snow rocky mountains peaks canyon. snow blanketed rocky mountains surround and shadow deep canyons. the canyons twist and bend through the high elevated mountain peaks, featuring a steady and smooth perspective.

Figure 6: Visualization results of multi-steps and an few-steps model.

compare our Stage I initialization with the learnable masked pruning strategy (Fang et al., 2025), which is very similar to the initialization method in Wu et al. (2025a), and our model under our Stage I

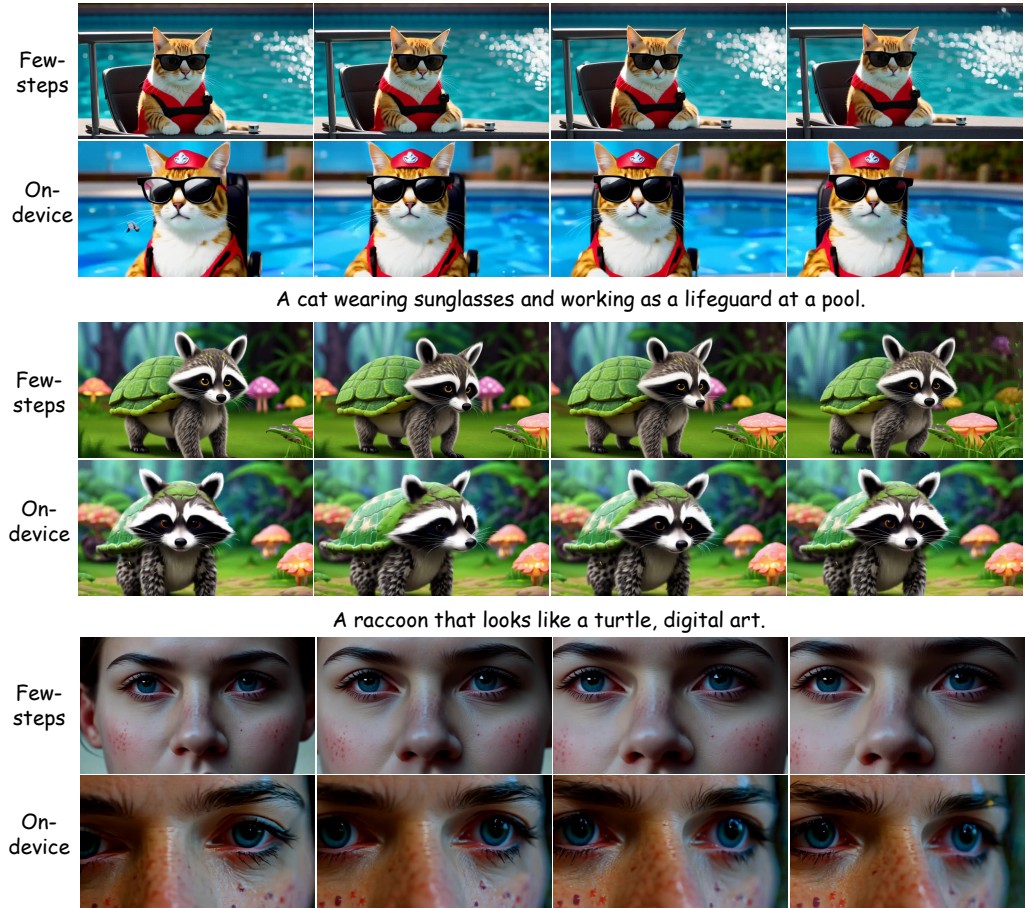

A cat wearing sunglasses and working as a lifeguard at a pool.

A raccoon that looks like a turtle, digital art.

The video begins with a close-up view of a person's face against a blurred, light-colored background, giving it a soft, artistic quality. As the camera zooms in, more details of the person's face become visible, focusing on the eyes and nose area. The eyes are a striking blue color, with defined eyelashes and eyebrows, and the skin shows reddish-brown spots or freckles. The nose is noted to be slightly crooked or asymmetrical. Throughout, the lighting remains natural, casting subtle shadows that accentuate the facial features. The sequence provides an intimate and detailed look at the person's facial features, particularly highlighting the blue eyes and the skin's imperfections.

Figure 7: Visualization results of on-device deployment and an A800 GPU server.

setting. Specifically, we prune the layers and the FFN intermediate dimensions of Wan2.1-1.3B (Wan et al., 2025), resulting in a 22-layer model with approximately 0.8B parameters, which is nearly the same as ours. We train both models for $10K$ iterations and evaluate them on Vbench (Huang et al., 2024), with the results presented in Table 5. The comparison shows that our initialization strategy achieves superior generation performance under the same recovery budgets.

## D    MORE VISUALIZATION RESULTS

We present more visualization results of our multi-steps, few-steps model as well as on-device few-steps model. The results is shown in Fig 6 and Fig 7.

## E    LIMITATIONS

Despite the promising results, our approach still has several limitations. First, the model occasionally struggles with fine-grained details, leading to noticeable distortions in object shapes (e.g., the keyboard in Fig. 8(a)). Second, due to limited training data and computational budget, our method can produce artifacts in human motion, such as limb deformities (e.g., the woman's hand in Fig. 8(b)). Finally,

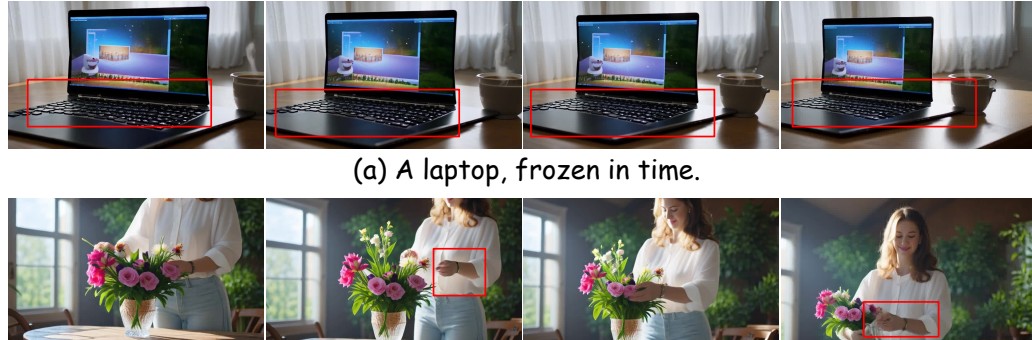

(a) A laptop, frozen in time.

(b)A person is arranging flowers.

Figure 8: Visualization of model limitations.

the evaluation is constrained by the lack of open-source implementations for certain baselines (e.g., SnapGen-V(Chen et al., 2025)), making it challenging to ensure strictly fair comparisons under identical settings.

