# OpenReview forum: "Lightning Video: Building Compact Diffusion Transformers for High-Fidelity On-Device Video Generation"
_ICLR.cc/2026/Conference — ICLR 2026 Conference Withdrawn Submission_

### Official Review · Reviewer_PZc9 · 2025-10-30

**Soundness:** 2
**Presentation:** 2
**Contribution:** 2
**Rating:** 4
**Confidence:** 3

**Summary:**

This paper introduce Lightning Video, a compact yet powerful DiT-based T2V model. This model aims at on-device video generation, and this approach combines (a) a Taylor-expansion-based pruning intialization to rescale a teacher DiT without backprop; (b) a two-stage training -- alignment + flow-matching and fine-tuning; (c) distribution-matching step distillation to reduce denoising steps. A 0.8B-parameter student reports VBench competitive with larger models, and a few-step variant is demonstrated on an iPhone 16 Pro Max.

**Strengths:**

1. Clear, modular framework (init → staged training → step distill) for compressing video DiTs while keeping quality; easy to adopt. No-backprop initialization via Taylor sensitivity is simple and empirically stronger than a masked-learnable baseline in Stage I (Table 5).
2. Competitive VBench at small scale: 0.8B few-step model reaches 83.77 and improves object/class and spatial metrics.

**Weaknesses:**

1. Novelty: The proposed pipeline relies primarily on established components (pruning + knowledge distillation/alignment + DMD-style step distillation). The absence of comparisons against stronger pruning and knowledge-transfer baselines limits confidence in the robustness and advancement of the proposed method.
2. Inconsistencies and clarity issues:
  a. The target few-step inference number N is inconsistently reported as both 3 and 4 steps throughout the manuscript. Specifically, Table 1 lists 4 steps, while lines 307-308 state 3 steps. This discrepancy needs clarification.
  b. The claim of "quantization-free" on-device execution contradicts the reported use of an INT4 quantized text encoder during deployment. Please reconcile this inconsistency in messaging and clarify the quantization strategy.
3. On-device latency concerns: The deployment results indicate a latency of 598 seconds for generating 480×832×81 frames on the target device, with even the optimized "on-device" configuration requiring 73 seconds—far from real-time or interactive performance. A more comprehensive breakdown of latency and energy consumption across individual components (text encoder, denoiser, VAE decoder) would be valuable. Additionally, latency comparisons on consumer-grade GPUs (e.g., NVIDIA RTX 4090, 5080) under identical settings should be included to benchmark against other methods.
4. Data transparency: The training dataset comprises a mixture of public and internally collected data totaling 3.5 million samples. However, critical details regarding data licensing, filtering protocols, deduplication procedures, and responsible use policies remain underspecified. Reproducibility is contingent upon whether these dataset components and processing pipelines will be publicly released.
5. Temporal consistency and generalization limitations: The limitations section acknowledges persistent artifacts in complex shapes and motions (e.g., hands, keyboards) and residual quality gaps. Challenges in maintaining temporal stability over extended durations and accurately modeling intricate human motion remain unresolved. Quantitative evaluation on long-form videos (≥10 seconds) is notably absent and should be included to assess temporal coherence.

**Questions:**

Please see weaknesses section

---

### Official Review · Reviewer_8pP7 · 2025-10-30

**Soundness:** 3
**Presentation:** 3
**Contribution:** 1
**Rating:** 2
**Confidence:** 3

**Summary:**

This paper presents Lightning Video, a 0.8B step distilled diffusion model pruned and distilled from Wav2.1 1.3B. The experiments show that the model can significantly reduce the parameter and inference time of the original model, enabling it to be deployed on-device.

**Strengths:**

1. The paper is well written and easy to follow.
2. Experiments demonstrate the effectiveness of the proposed pipeline.

**Weaknesses:**

Major Weaknesses
● This paper is more like a A+B work. In detail:
  ○ The idea of using taylor expansion and hessian matrix to analyze the importance of parameters has already been well studied.
  ○ The proposed multi-stage training lacks salient motivation and seems trivial.
  ○ The step distillation is not significantly different from the DMD framework.
●  The main speed acceleration comes from step distillation, which is similar to existing DMD framework. The paper itself does not contribute new innovation to the acceleration of diffusion models.
● The parameter efficiency is not very significant. It only reduces the parameters from 1.3B to 0.8B. Therefore, the memory saved is also moderate.
● As an acceleration method, it lacks comparison to other model acceleration methods. For example, parameter quantization can save more memory than this work, while also maintaining similar generation performance.

Minor Weaknesses
● The title of Figure 1 claimed that Figure 1(b) contains running time comparison on mobile device, which is not shown.
● Figure 2 can be improved to better illustrate which parts correspond to each step of the proposed pipeline.
● Some punctuation marks are used incorrectly. For example, there should be a period after equation (2) and the comma after equation (4) should also be a period.
● Some references are not well formatted and contain duplicated reference. For example, references in L665 and L669 refer to the same paper.
● There are many typos in the paper. For example, in the title of Section 2.1, it shall be "TAYLOR-BASED" rather than "TAYLER-BASED".

**Questions:**

● What is the difference of this paper with existing works? Specifically,
  ○ Does it provide more insight in analyzing the taylor expansion and hessian matrix than existing works?
  ○ What is the motivation of the multi-stage training in Sec. 2.2?
  ○ What is the difference between the proposed step distillation with the DMD framework?
● Does this paper introduce new innovation in accelerating diffusion models?
●  What is the memory used before and after applying the proposed steps? How does it compare to parameter quantization approaches? In detail:
  ○ When the performance are similar, can it save more memory than parameter quantization approaches?
  ○ When fairly equipped w/wo step distillation, does the proposed model run faster or generates better than parameter quantization approaches?

---

### Official Review · Reviewer_fdEP · 2025-11-01

**Soundness:** 3
**Presentation:** 3
**Contribution:** 3
**Rating:** 6
**Confidence:** 4

**Summary:**

This paper presents an innovative compressed diffusion transformer model that addresses the challenge of deploying large-scale video generation models on mobile devices. The innovation lies in three key components: Taylor expansion-based pruning initialization, a staged training protocol, and distribution matching distillation strategy. Experimental results show that, with 0.8B parameters and medium-to-low resolution, the model can achieve 16fps video generation on mobile devices.

**Strengths:**

1.The paper proposes a model acceleration method for DiT (Diffusion Transformer) in mobile applications. The fusion of pruning initialization, staged training, and distribution matching distillation, especially the Taylor expansion-based pruning initialization, effectively avoids the instability issues in previous initializations.

2.The experimental design is comprehensive, and the arguments are well-supported. The writing is clear and easy to understand, with well-defined concepts.

**Weaknesses:**

1.Although the model performance is improved through pruning and knowledge distillation, the training and distillation process requires substantial computational power and time, making training the model more difficult, especially the distribution matching distillation. This may limit the model's widespread adoption.

2.There are significant differences in memory bandwidth and computational capabilities across different edge devices, which could affect the model's performance. While good performance is shown on the iPhone 16 Pro, the paper could explore performance on various devices, particularly lower-end configurations.

3.The speed improvement may primarily stem from DMD distillation, which presents limited innovation. The paper lacks ablation studies regarding the distillation component.

**Questions:**

1.Were the time and memory costs of the two-stage training and distribution matching distillation strategy, as mentioned in Lighting Video, thoroughly evaluated in the implementation? The training cost of these steps is critical for reproducibility.

2.Did the paper explore the impact of different pruning strategies on model performance? For example, were there experiments comparing the performance differences under various pruning ratios or different pruning strategies?

---

### Official Review · Reviewer_mQKE · 2025-11-02

**Soundness:** 2
**Presentation:** 3
**Contribution:** 2
**Rating:** 4
**Confidence:** 5

**Summary:**

The paper starts from Wan2.1-1.3B and proposes (1) a Taylor-based pruning initialization, (2) multi-level alignment followed with high-quality data fine-tuning, and (3) distribution matching distillation for few-step inference. The final 0.8B DiT claims VBench is on par with the large-scale model and achieves on-device inference on the iPhone 16 Pro Max for ~70s per sample.

**Strengths:**

- The paper proposed a Taylor-based pruning method, which only requires a forward pass for pruning initialization.
- The final model achieves on-par quality with its base model.

**Weaknesses:**

- The motivation is not fully convincing. The paper targets a 0.8B DiT for on-device video generation. However, the reported inference latency remains ~70 seconds, which is too slow for practical mobile use and will likely result in significant energy consumption. This weakens the claim of “on-device” feasibility.
- The proposed weight importance estimation is trivial, as a similar Taylor-based weight importance estimation has been explored in [1], which diminishes the contribution of this component.
- Missing the experiment of showing that the 1.3B baseline cannot be deployed on-device with engineering. For example, Apple's CoreML supports model chunking (i.e., `ct.models.utils.bisect_model`), a practical approach for fitting large models on-device. Comparing against such deployment-oriented baselines would strengthen the claim.

[1] Molchanov, Pavlo, et al. "Importance estimation for neural network pruning." Proceedings of the IEEE/CVF conference on computer vision and pattern recognition. 2019.

[2] Fang, Gongfan, et al. "Tinyfusion: Diffusion transformers learned shallow." Proceedings of the Computer Vision and Pattern Recognition Conference. 2025.

**Questions:**

- What is the on-device latency breakdown across components, e.g., text encoder, VAE/decoder, and the DiT?
- Pruning configuration:
  - Why was the 850M model size chosen?
  - Did you use a uniform pruning ratio for all layers?
  - Since the method only needs forward passes for initialization, is there a reason you did not adopt non-uniform (layer-/module-specific) pruning configurations?
- How many inference steps? line-307 states `3 inference steps` while Tab.1 states `4 steps`.

- typo:
  - fig.1 `satge2` --> `stage 2`
  - fig.2 `Tayler` --> `Taylor`

---

### Note · Authors · 2025-11-14

I have read and agree with the venue's withdrawal policy on behalf of myself and my co-authors.